# APPLICATION OF ARTIFICIAL INTELLIGENCE TO PREDICT VIBRATIONS IN MECHANICAL SYSTEMS WITH MOVING BOUNDARIES

**Vladislav L. Litvinov**
Samara State Technical University
Samara, 446001, Russia
vladlitvinov@rambler.ru

## Abstract

The study of the dynamics of objects with moving boundaries described by one-dimensional boundary value problems is of significant interest for modern engineering and scientific calculations. Of particular relevance is the analysis of the resonant properties of objects capable of causing critical states and structural failure. The article presents the developed software package "TB-ANALYSIS", created in the MATLAB environment for mathematical modeling and analysis of the resonant characteristics of objects with moving boundaries. This package is designed to solve one-dimensional boundary value problems describing the dynamics of objects of variable length. The main objectives of the work are: creating software for studying solutions to boundary value problems and modeling resonance phenomena; testing the effectiveness of the software package; developing an algorithmic description of the functionality. A key feature of the package is the integration of classical numerical methods with modern artificial intelligence algorithms. To solve boundary value problems, three main methods are implemented: an analytical method for replacing variables in a system of functional-difference equations, an asymptotic method, and an approximate method for constructing solutions to integro-differential equations. Intelligent selection of the method is carried out automatically depending on the type of the problem being solved. The system's architecture includes interconnected modules accessible through a single graphical interface. The main modules are designed to investigate solutions to boundary value problems, analyze resonance properties, and control resonance phenomena. To ensure calculation accuracy, a special error estimation procedure is implemented at each step. To demonstrate the system's capabilities, a study was conducted on the transverse vibrations of a variable-length viscoelastic cable resting on an elastic foundation. The results of a comparative analysis confirmed the effectiveness of the numerical methods and the correct operation of the algorithms. The study focuses on the use of artificial intelligence, neural networks, and machine learning to analyze resonance phenomena, predict optimal system parameters, and prevent resonance. The use of deep neural networks and adaptive control increased the accuracy of predictions and the effectiveness of system control. As a result, the neural network can predict resonant frequencies and suggest optimal parameters. Calculations confirmed that the parameters suggested by the artificial intelligence do indeed prevent resonance. A comparative analysis conducted across multiple test cases demonstrated that the developed hybrid approach reduces the resonant frequency prediction error to 2.1% and prevents resonance in 96% of cases, which is 34% more accurate and, on average, 5 times faster than classical methods. The developed software package is an effective tool for studying the dynamics of objects with moving boundaries. The key advantages of the package include: versatility for solving various classes of boundary value problems; intelligent selection of solution methods; a user-friendly interface; and built-in tools for evaluating computational accuracy. Prospects for further development include expanding the class of problems solved and implementing additional numerical methods.

## 1 INTRODUCTION

Objects with moving boundaries are widely used in various technical devices, such as ropes in

lifting mechanisms [1–5, 7, 8, 13] or flexible elements in transmissions [6]. The main difficulty in their mathematical description is associated with taking into account the moving boundaries. Exact solution methods are limited by the wave equation and relatively simple boundary conditions [9]. Among the approximate methods, the most effective are the method for constructing solutions of integro-differential equations, described in [7, 11, 21, 22], as well as the Kantorovich–Galerkin method [8–10, 14]. These methods have been adapted for more complex models that take into account bending rigidity, the influence of the external environment, and the elastic foundation of the vibrating object.

This article presents the development of the TB-ANALYSIS software package [23], created in the MATLAB environment for mathematical modeling and analysis of the resonant characteristics of objects with moving boundaries, described by one-dimensional boundary value problems. The formal statement of the problem to be solved is presented in Section 2. Three main methods are implemented to solve boundary value problems: the analytical method of replacing variables in a system of functional-difference equations [16], the asymptotic method [12], and the approximate method of constructing solutions of integro-differential equations [21, 22]. Intelligent selection of the method is performed automatically depending on the type of the problem to be solved (class of integro-differential or differential equation, initial and boundary conditions, etc.). A key feature of the package is the hybrid architecture described in Section 3, which integrates classical numerical methods with machine learning algorithms into a single iterative workflow. Key functional capabilities of the software package include: obtaining and analyzing numerical solutions of model equations for various system parameters and external influences; studying the influence of multiple factors on resonance properties, including the type of object, boundary velocity, oscillation modes, drag forces, viscoelastic characteristics, bending stiffness and substrate rigidity; Optimization of model parameters to eliminate unwanted resonance phenomena. A key feature of this software package is the resonance management function, implemented within the analysis and adaptive control module. Its purpose is to automatically determine the frequency range in which resonance occurs and create conditions for its elimination by optimizing model parameters.

Using artificial intelligence, the system's resonant frequencies were determined, as well as the conditions under which resonance can be prevented. Key parameters influencing resonance were identified. The hybrid neural network ResonanceNet (Section 7) was trained using this data, enabling it to predict dangerous resonant frequencies and propose optimal parameters. To quantitatively evaluate the advantages of the developed hybrid approach, Section 8 compares its accuracy and performance with classical methods. Thus, the proposed approach based on AI and machine learning demonstrates improved accuracy and reduced computation time. Modern research also confirms the effectiveness of hybrid approaches combining physical models with deep learning for predicting and suppressing resonant phenomena in dynamic systems [28–30].

## 2. STATEMENT OF THE PROBLEM

We consider a class of mechanical systems with moving boundaries, the dynamics of which are described by boundary value problems for partial differential equations of the form:

$$U_{\tau\tau}(\xi,\tau) + L[U(\xi,\tau)] = \varphi\big(\xi,\tau,U(\xi,\tau),U_{\xi}(\xi,\tau)\big), \xi \in [0,l(\varepsilon\tau)], \tau > 0,$$

where $U(\xi,\tau)$ is the desired function (transverse or longitudinal displacement); $L$ is a linear differential or integro-differential operator that takes into account the elastic and viscoelastic properties of the object, the resistance of the medium, and the rigidity of the substrate; $l(\varepsilon\tau)$ is the specified law of change in the length of the object (boundary motion); $\varepsilon$ is a small parameter characterizing the slowness of boundary motion; $\varphi$ is a specified function of class $C$, characterizing the external influence.

Boundary conditions are specified as follows:

$$Y_{ji}\Big[U\big(l_j(\varepsilon\tau),\tau\big)\Big] = F_{ji}(\tau);$$

$$i = \overline{1,m}; \ j = \overline{1,2}.$$

Here $Y_{ji}$ are linear homogeneous differential operators with respect to $\xi$, $m \le 2$ is a positive

integer; $F_{ji}(\tau)$ are given functions of class $C^2$.

When analyzing resonance properties, the initial conditions are assumed to be zero.
The main objectives of the work are:
1. Develop a hybrid numerical-analytical method combining classical approaches (asymptotic method, Kantorovich-Galerkin method, method of constructing solutions of integro-differential equations) with machine learning algorithms for predicting resonance characteristics.
2. Based on this, create the "TB-ANALYSIS" software package, which provides:
– Automatic selection and application of the optimal classical method for generating training data and verification.
– Prediction of resonant frequencies and amplitudes using a specialized neural network.
– Adaptive control of system parameters to suppress unwanted resonant modes.
3. Conduct a comparative quantitative analysis of the accuracy and effectiveness of the hybrid approach compared to classical methods.

## 3. SOFTWARE ARCHITECTURE

The TB-ANALYSIS software suite's modular architecture implements an iterative hybrid algorithm that combines classical numerical methods with machine learning. The main modules interact according to the following scheme:
3.1. Parameterization and preliminary numerical modeling module
The system parameters and boundary conditions are input. Based on an analysis of the problem type (equation type, boundary conditions), one of three classical methods (analytical, asymptotic, and approximate) is automatically selected. The selected method is used for rapid calculations over a limited time interval and the formation of an initial feature vector $X_0$, which can also be supplemented for subsequent training of the neural network.
3.2. Machine learning module "ResonanceNet"
The feature vector $X_0$ is input to a pre-trained hybrid neural network. The network performs multi-task regression, producing a prediction of the target variables $y_{pred} = y(f_{res}, A_{max}, R_{risk})$, where $f_{res}$ is the estimated resonant frequency, $A_{max}$ is the estimated amplitude, and $R_{risk}$ is the resonance risk category.
3.3. Analysis, verification and adaptive control module
Forecast $y_{pred}$ is analyzed for reliability. If the network's confidence level (measured through the prediction variance in Monte Carlo dropout) is below a specified threshold, control is transferred to the advanced numerical simulation module for a high-precision calculation using the classical method. The obtained results are added to the database for further network training. Based on the final forecast, recommendations are generated for adjusting system parameters (stiffness, damping) to shift the operating point away from the resonant region.
3.4. Visualization and Results Output Module
Provides tools for graphical and tabular presentation of results, including plotting dependencies, comparing methods, and generating final reports. This architecture effectively combines the prediction speed of AI methods with the accuracy and reliability of classical numerical methods, creating a self-learning system for analyzing the dynamics of mechanical systems with moving boundaries.

## 4. ANALYSIS OF RESONANCE PROPERTIES OF MODELS USING A SOFTWARE PACKAGE

Activating the "Analysis of Resonance Properties of Models" button in the start window initiates a transition to a specialized interface that maintains an identical menu structure to the interface of the boundary value problem solution research module.
The module's interface includes control panels for configuring study parameters: the "Study Object" switch allows selection between rope and beam models, while the "Dependency Analysis" panel provides a choice between two study types: amplitude time dependence analysis and the dependence of maximum amplitude on boundary velocity. Computational procedures are implemented through a system of built-in functions: the calculation of amplitude-

time characteristics is performed by the "met_ampl" function, while the analysis of the maximum amplitude's dependence on velocity is performed by the "met_ampl_max" function, which uses "met_ampl" as a subroutine. This block also provides the ability to vary key model parameters, including the mode number, damping coefficients, object stiffness, viscoelastic properties, and substrate stiffness, using the corresponding input fields.

The interactive "Calculate" button is used to initialize the calculation procedures and subsequently visualize the resulting dependencies in graphical format.

The dependence of maximum amplitude on speed is determined using an original method developed as part of this study. The method is based on an analytical expression for oscillation amplitude, derived through solving integro-differential equations, taking into account the resonant characteristics of the mechanical systems under study. The amplitude of the system's oscillations in the nth dynamic mode is described by the following expression:

$$A_n^2(\tau) = E_n^2(\tau)\left\{\left[\int_0^\tau F_n(\zeta)\cos\Phi_n(\zeta)d\zeta\right]^2 + \left[\int_0^\tau F_n(\zeta)\sin\Phi_n(\zeta)d\zeta\right]^2\right\}. \tag{1}$$

The algorithm for numerically studying steady-state resonance and the phenomenon of passing through resonance is implemented in the "met_ampl_max" function. Figure 1 shows a graph of the dependence of the maximum amplitude of rope oscillations when passing through resonance on the boundary velocity for various dimensionless values of the medium resistance coefficient (from top to bottom: $\alpha = 0$; $\alpha = 0,01$) with the following model parameters: mode number 1; object stiffness coefficient 0.01; viscoelasticity coefficient 0.01; substrate stiffness coefficient 0.02.

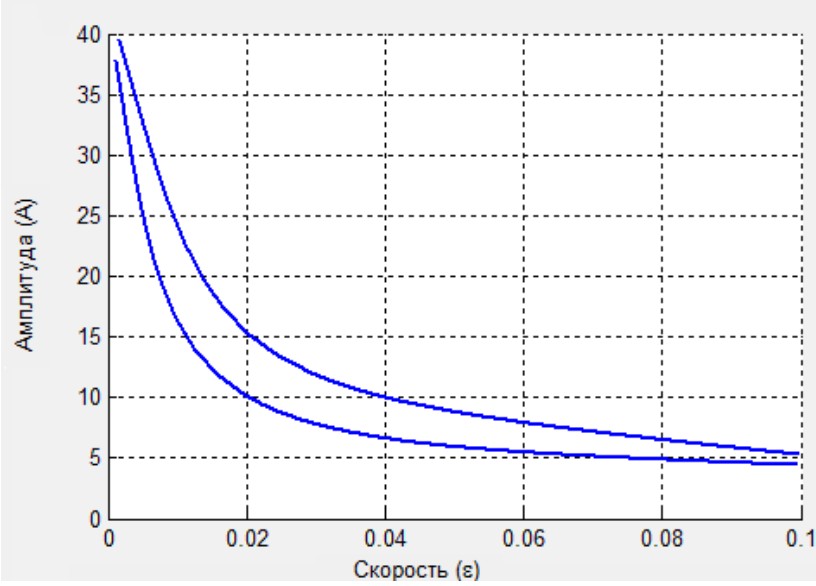

Fig. 1. Graph of the dependence of the maximum amplitude on the velocity of boundary movement for different values of the medium resistance coefficient (from top to bottom: $\alpha = 0$; $\alpha = 0,01$).

The calculation of the time dependence of the oscillation amplitude according to formula (1) is implemented in the internal function "met_ampl", which is used as a subroutine in the function "met_ampl_max". Figure 2 presents the results of test calculations demonstrating the change in the amplitude of transverse oscillations of a variable-length rope when passing through resonance in the first dynamic mode for a specific set of initial model parameters, implemented in dimensionless variables.

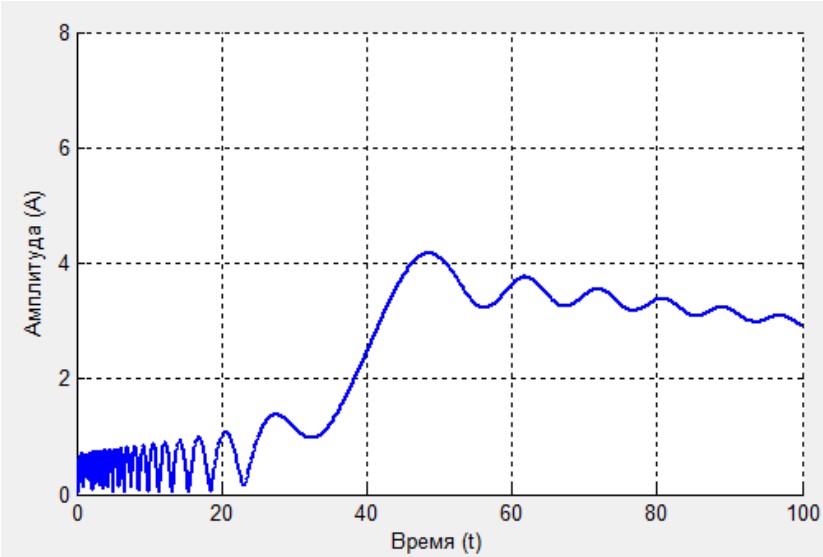

Fig. 2. Graph of amplitude versus time as it passes through resonance.

In addition to their direct functional role, the graphical dependencies shown in Figures 1 and 2 clearly demonstrate the characteristic features of the behavior of the oscillation amplitude, which form the basis of the method for determining the maximum amplitude [9, 16].

## 5. ESTIMATION OF THE ERROR IN CALCULATING THE AMPLITUDE OF OSCILLATIONS

Let us move on to the assessment of the error in calculating the square of the amplitude [16]

$$A_n^2(\tau_1, \tau_2) = E_n^2(\varepsilon\tau_2)\left\{\left[\int_{\tau_1}^{\tau_2} F_n(\varepsilon\zeta)\cos\Phi_n(\zeta)d\zeta\right]^2 + \left[\int_{\tau_1}^{\tau_2} F_n(\varepsilon\zeta)\sin\Phi_n(\zeta)d\zeta\right]^2\right\} \quad (2)$$

when the system passes through resonance. In [21] an important auxiliary lemma was proved:
Lemma 1. Adding and subtracting a constant to a function $z(\tau)$ does not affect the result of calculating the function.

$$g(\varepsilon, \tau_1, \tau_2) = \left[\int_{\tau_1}^{\tau_2} f(\zeta)\cos[z(\zeta)]d\zeta\right]^2 + \left[\int_{\tau_1}^{\tau_2} f(\zeta)\sin[z(\zeta)]d\zeta\right]^2.$$

According to Lemma 1, the error in the numerical calculation of function $\Phi_n(\zeta)$ does not affect the accuracy of expression (2), which significantly simplifies the procedure for estimating the total error. Thus, the reliability of the amplitude calculation using expression (2) is determined solely by the accuracy of the calculation of the two integrals:

$$I_1 = \int_{\tau_1}^{\tau_2} F_n(\varepsilon\zeta)\cos\Phi_n(\zeta)d\zeta, \quad I_2 = \int_{\tau_1}^{\tau_2} F_n(\varepsilon\zeta)\sin\Phi_n(\zeta)d\zeta. \quad (3)$$

Let us define the maximum error in calculating the integrals as $R_{int} = \max\left\{R_{int}^1, R_{int}^2\right\}$, where $R_{int}^1, R_{int}^2$ are the errors in calculating the integrals $I_1, I_2$, respectively
Then expression (2) can be represented as follows:

$$A_n^2(\tau_1, \tau_2) = E_n^2(\tau_2)\left\{\tilde{I}_1^2 + \tilde{I}_2^2\right\}, \quad (4)$$

where $\tilde{I}_1 = I_1 + R_{int}$ is the approximate value of $I_1$, and $\tilde{I}_2 = I_2 + R_{int}$ is the approximate value of $I_2$. Let us substitute expressions for approximate values into (4):

$$\tilde{A}_n^2(\tau_1, \tau_2) = E_n^2(\tau_2)\left\{\left(I_1 + R_{int}\right)^2 + \left(I_2 + R_{int}\right)^2\right\} =$$

$$= E_n^2(\tau_2)\left(I_1^2 + I_2^2\right) + 2E_n^2(\tau_2)R_{int}\left[R_{int} + I_1 + I_2\right] =$$
$$= A_n^2(\tau_1, \tau_2) + 2E_n^2(\tau_2)R_{int}\left[R_{int} + I_1 + I_2\right].$$

From here, the error in calculating the square of the amplitude (2) is calculated using the formula

$$R = 2E_n^2(\tau_2)R_{int}\left[R_{int} + I_1 + I_2\right]. \tag{5}$$

The software package implements a step-by-step error estimation algorithm based on formula (5). Calculation accuracy is controlled by adjusting the calculation accuracy of integrals (3), using the built-in MATLAB quad function, designed for numerical integration and allowing the required calculation accuracy to be specified via the corresponding input parameter.

An additional factor for calculation optimization is the use of the additivity property of the integral operator, which significantly reduces calculation time:

$$\int_{\tau_1}^{\tau_2} F_n(\zeta)\cos\Phi_n(\zeta)d\zeta = I_1,$$

$$\int_{\tau_1}^{\tau_2 + \triangle\tau} F_n(\zeta)\cos\Phi_n(\zeta)d\zeta = I_1 + \int_{\tau_2}^{\tau_2 + \triangle\tau} F_n(\zeta)\cos\Phi_n(\zeta)d\zeta,$$

where $\triangle\tau > 0$ is the step along $\tau$.

## 6. ESTIMATION OF THE ERROR IN CALCULATING THE TRANSVERSE VIBRATIONS OF A VISCOELASTIC ROPE OF VARIABLE LENGTH

### 6.1 Statement of the problem

The problem under consideration of analyzing the transverse vibrations of a viscoelastic rope of variable length, placed on an elastic foundation and possessing bending rigidity while taking into account the resistance forces of the environment, is a special case of the general formulation of the problem [9].

The differential equation describing the vibrations of the rope is:

$$U_{tt}(x,t) - a^2 U_{xx}(x,t) + \frac{\lambda}{\rho}U_t(x,t) + \frac{EI}{\rho}U_{xxxx}(x,t) + \frac{\mu I}{\rho}U_{xxxxt}(x,t) + \frac{k_0}{\rho}U(x,t) = 0. \tag{6}$$

The boundary conditions can be written as follows:

$$U(0,t) = 0; \; U_{xx}(0,t) = 0; \tag{7}$$

$$U(l_0(t),t) = B\cos W_0(\omega_0 t); \; U_x(l_0(t),t) = 0. \tag{8}$$

In problem (6) – (8) the following notations are used:

$U(x,t)$ is the transverse displacement of a point on the rope with coordinate x at time t; $E$ is the modulus of elasticity of the rope material; I is the axial moment of inertia of the rope cross-section; $\mu$ is the parameter characterizing the viscoelasticity of the object; $\lambda$ is the resistance force of the medium acting on a unit length of the rope at a unit velocity of transverse motion; $k_0$ is the rigidity of the substrate; $a = \sqrt{T/\rho}$ is the minimum velocity of wave propagation, T is the tension force, $\rho$ is the linear mass density of the rope; $l_0(t) = L_0 - v_0 t$ is the law of motion of the rope boundary; $L_0$ is the initial length of the rope; $W_0(z)$ is a function of class $C^2$; $B, \omega_0$ are constant values.

Let us introduce dimensionless variables into problem (6) – (8):

$$\xi = \omega_0 x / a; \; \tau = \omega_0 t + \frac{\omega_0 L_0 - a}{-v_0}; \; U(x,t) = Bu(\xi, \tau),$$

and a new function $u(\xi, \tau) = e^{-\alpha\tau}V(\xi, \tau)$, where $\alpha = \lambda / (2\omega_0 \rho)$.

Then problem (6) – (8) will take the form:

$$V_{\tau\tau}(\xi, \tau) - V_{\xi\xi}(\xi, \tau) - \sigma^2 V(\xi, \tau) + (\beta^2 - \alpha\gamma^2)V_{\xi\xi\xi\xi}(\xi, \tau) + \gamma^2 V_{\xi\xi\xi\xi\tau}(\xi, \tau) = 0; \tag{9}$$

$$V(0,\tau) = 0; \; V_{\xi\xi}(0,\tau) = 0; \tag{10}$$

$$V(l(\varepsilon\tau),\tau) = e^{\alpha\tau}\cos W(\tau); \; V_\xi(l(\varepsilon\tau),\tau) = 0, \tag{11}$$

where

$$\beta^2 = \frac{EI}{\rho}\frac{\omega_0^2}{a^4}; \quad \gamma^2 = \frac{\mu I}{\rho}\frac{\omega_0^3}{a^4}; \quad l(\varepsilon\tau) = 1 + \varepsilon\tau; \quad \sigma^2 = \alpha^2 - \eta;$$

$$\eta = \frac{k_0}{\rho\omega_0^2}; \quad W(\tau) = W_0(\tau - \gamma_0); \quad \gamma_0 = \frac{\omega_0 L_0 - a}{-v_0}; \quad \varepsilon = -v_0 / a.$$

6.2 Solution to the problem

We will solve problem (9) – (11) in dimensionless variables in accordance with the methodology described in [8, 9, 14–16].

Let us introduce the notations $\delta^2 = (\beta^2 - \alpha\gamma^2)$ and $\Omega_{0n}^2(\varepsilon\tau) = \omega_{0n}^2(\varepsilon\tau) - \sigma^2$, where $\omega_{0n}(\varepsilon\tau)$ are the natural frequencies of problem (9) – (11).

Let us find an expression for the dynamic modes $X_n(\xi,\varepsilon\tau)$ and functions $\omega_{0n}(\varepsilon\tau)$:

$$X_n(\xi,\varepsilon\tau) = A_n\{\sin[k_1(\varepsilon\tau)\xi] + c_n(\varepsilon\tau)sh[k_2(\varepsilon\tau)\xi]\};$$

$$\omega_{0n}(\varepsilon\tau) = [\omega_{1n}(\varepsilon\tau) + d_n(\varepsilon\tau)]\sqrt{1 + \delta^2[\omega_{1n}(\varepsilon\tau) + d_n(\varepsilon\tau)]^2},$$

where

$$A_n = 1 / \max\{\sin[k_1(\varepsilon\tau)\xi] + c_n(\varepsilon\tau)sh[k_2(\varepsilon\tau)\xi]\};$$

$$k_1(\varepsilon\tau) = \frac{1}{\sqrt{2}\delta}\sqrt{-1 + \sqrt{1 + 4\delta^2\omega_{0n}^2(\varepsilon\tau)}}; \quad k_2(\varepsilon\tau) = \frac{1}{\sqrt{2}\delta}\sqrt{1 + \sqrt{1 + 4\delta^2\omega_{0n}^2(\varepsilon\tau)}};$$

$$c_n(\varepsilon\tau) = -\frac{\sin[k_1(\varepsilon\tau)l(\varepsilon\tau)]}{sh[k_2(\varepsilon\tau)l(\varepsilon\tau)]}; \quad \omega_{1n}(\varepsilon\tau) = \frac{\pi n}{l(\varepsilon\tau)}; \quad d_n(\varepsilon\tau) = \frac{1}{l(\varepsilon\tau)}arctg\frac{\delta\omega_{1n}(\varepsilon\tau)}{\sqrt{1 + \delta^2\omega_{1n}^2(\varepsilon\tau)}}.$$

We will look for a solution to the problem in the form

$$V(\xi,\tau) = \sum_{n=1}^{\infty}\mu_n(\tau)X_n(\xi,\varepsilon\tau),$$

where $\mu_n(\tau) = A_{0n}(\varepsilon\tau)y_n(\tau)$, and the function $y_n(\tau)$ satisfies the following equation, written with an accuracy of up to values of the order of $\varepsilon^2$:

$$y_n''(\tau) + \Omega_{0n}^2(\varepsilon\tau)y_n(\tau) = -\frac{\omega_{0n}^2(\varepsilon\tau)Q_{n_{21}}(\varepsilon\tau)}{A_{0n}(\varepsilon\tau)}e^{\alpha\tau}\cos W(\tau). \tag{12}$$

Making calculations for functions $Q_{n_{21}}(\varepsilon\tau)$, $A_{0n}$ we obtain:

$$Q_{n_{21}}(\varepsilon\tau) = \frac{-k_1(\varepsilon\tau)\sqrt{1 + 4\delta^2\omega_{0n}^2(\varepsilon\tau)}}{\omega_{0n}^2(\varepsilon\tau)A_{1n}(\varepsilon\tau)}\cos[k_1(\varepsilon\tau)l(\varepsilon\tau)];$$

$$A_{0n}(\varepsilon\tau) = 1/\sqrt{A_{1n}(\varepsilon\tau)}; \quad A_{1n}(\varepsilon\tau) = \frac{1}{2}l(\varepsilon\tau)[1 - c_n^2(\varepsilon\tau)] - \frac{\sin[2k_1(\varepsilon\tau)l(\varepsilon\tau)]}{4k_2(\varepsilon\tau)\omega_{0n}(\varepsilon\tau)\delta}.$$

Using the asymptotic method and the small parameter method [12], we find that two linearly independent solutions of the homogeneous equation corresponding to (12) have the form:

$$y_{1n}(\tau) = a_n(\varepsilon\tau)\cos w_n(\tau); \quad y_{2n}(\tau) = a_n(\varepsilon\tau)\sin w_n(\tau),$$

where

$$a_n(\varepsilon\tau) = 1/\sqrt{\Omega_{0n}(\varepsilon\tau)}; \quad w_n(\tau) = \int_0^\tau \Omega_{0n}(\varepsilon\zeta)d\zeta.$$

The oscillation amplitude corresponding to the n-th dynamic mode has the form (1), where

$$E_n^2(\varepsilon\tau) = \frac{e^{-2\alpha\tau}}{4A_{1n}(\varepsilon\tau)\Omega_{0n}(\varepsilon\tau)}; \quad \Phi_n(\zeta) = w_n(\zeta) - W_n(\zeta);$$

$$F_n(\varepsilon\zeta) = \omega_{0n}^2(\varepsilon\zeta)Q_{n_{21}}(\varepsilon\zeta)e^{\alpha\zeta}\sqrt{A_{1n}(\varepsilon\zeta)/\Omega_{0n}(\varepsilon\zeta)}.$$

6.3 Analysis of resonance properties

Using the methodology described in [9], we will consider the phenomenon of steady-state

resonance and passage through resonance.

Steady-state resonance in the system under consideration is observed if

$$W_n(\tau) = w_n(\tau) + \gamma_1,$$

where $\gamma_1$ is a constant value. The amplitude in this case has the form

$$A_n(\tau) = E_n(\varepsilon\tau) \int_0^\tau F_n(\varepsilon\zeta) d\zeta.$$

When a harmonic disturbance with frequency $\omega_0$, acts on the system when $W(\tau) = \tau$, the phenomenon of passing through resonance may occur on any of the dynamic modes.

The point of the resonant region $\tau_0$, in which $\Phi_n'(\tau_0) = 0$, is approximately determined by the following formula:

$$\tau_0 = \frac{1}{\varepsilon}\left[ \sqrt{\frac{2\delta^2}{-1+\sqrt{1+4\delta^2(1+\sigma^2)}}} \cdot \pi n - 1 \right].$$

The expression for the maximum possible amplitude when passing through resonance has the form (2).

6.4 Estimation of calculation error

Let's estimate the calculation error.

The solution to problem (6–8) is as follows:

$$u(\xi,\tau) = e^{-\alpha\tau} \sum_{n=1}^{\infty} \mu_n(\tau) X_n(\xi,\varepsilon\tau). \tag{13}$$

By discarding the terms of the series starting from the k-th in (13), we obtain an expression for the error in the form

$$R_k = e^{-\alpha\tau} \sum_{n=k}^{\infty} \mu_n(\tau) X_n(\xi,\varepsilon\tau). \tag{14}$$

To calculate (14) it is necessary to find the dependence of the error on the number k. At $\alpha > 0$, $\tau > 0$, we have

$$|R_k| = \left| e^{-\alpha\tau} \sum_{n=k}^{\infty} \mu_n(\tau) X_n(\xi,\varepsilon\tau) \right| \le \left| \sum_{n=k}^{\infty} \mu_n(\tau) X_n(\xi,\varepsilon\tau) \right| \le \sum_{n=k}^{\infty} \left| \mu_n(\tau) X_n(\xi,\varepsilon\tau) \right| =$$

$$= \sum_{n=k}^{\infty} |\mu_n(\tau)| |X_n(\xi,\varepsilon\tau)|,$$

where the amplitude modulus $\mu_n(\tau)$ is determined by the expression

$$|\mu_n(\tau)| = E_n(\varepsilon\tau) \sqrt{\left\{ \left[ \int_0^\tau F_n(\varepsilon\zeta) \cos\Phi_n(\zeta) d\zeta \right]^2 + \left[ \int_0^\tau F_n(\varepsilon\zeta) \sin\Phi_n(\zeta) d\zeta \right]^2 \right\}}.$$

The error in calculating integrals $I_1 = \int_{\tau_1}^{\tau_2} F_n(\varepsilon\zeta) \cos\Phi_n(\zeta) d\zeta$, $I_2 = \int_{\tau_1}^{\tau_2} F_n(\varepsilon\zeta) \sin\Phi_n(\zeta) d\zeta$ is

discussed in section 5.

Let's evaluate the factor $E_n(\varepsilon\tau) = \dfrac{e^{-\alpha\tau}}{\sqrt{4A_{1n}(\varepsilon\tau)\sqrt{\omega_{0n}^2 - \sigma^2}}}$ .

To evaluate $E_n(\varepsilon\tau)$, it is necessary to evaluate the components of $d_n(\varepsilon\tau), \omega_{0n}(\varepsilon\tau), A_{1n}(\varepsilon\tau), k_1(\varepsilon\tau), k_2(\varepsilon\tau), c_n(\varepsilon\tau)$.

Let's evaluate $d_n(\varepsilon\tau)$.

$$d_n(\varepsilon\tau) = \frac{1}{l(\varepsilon\tau)} arctg \frac{\delta \dfrac{\pi n}{l(\varepsilon\tau)}}{\sqrt{1 + \delta^2 \dfrac{\pi^2 n^2}{l^2(\varepsilon\tau)}}} = \frac{1}{l(\varepsilon\tau)} arctg \frac{\delta\pi}{\sqrt{\dfrac{l^2(\varepsilon\tau)}{n^2} + \delta^2\pi^2}} .$$

$$\left.\frac{1}{l(\varepsilon\tau)}\,arctg\,\frac{\delta\pi}{\sqrt{\frac{l^2(\varepsilon\tau)}{n^2}+\delta^2\pi^2}}\right|_{n=1}\leq d_n(\varepsilon\tau)\leq\lim_{n\to\infty}\frac{1}{l(\varepsilon\tau)}\,arctg\,\frac{\delta\pi}{\sqrt{\frac{l^2(\varepsilon\tau)}{n^2}+\delta^2\pi^2}}\,.$$

$$\frac{1}{l(\varepsilon\tau)}\,arctg\,\frac{\delta\pi}{\sqrt{l^2(\varepsilon\tau)+\delta^2\pi^2}}\leq d_n(\varepsilon\tau)\leq\frac{1}{l(\varepsilon\tau)}\,arctg1\,.$$

As a result, we obtain an estimate for $d_n(\varepsilon\tau)$:

$$0<d_n(\varepsilon\tau)\leq\frac{\pi}{4l(\varepsilon\tau)}\,. \tag{15}$$

Now let's evaluate $\omega_{0n}(\varepsilon\tau)$.

$$\omega_{0n}(\varepsilon\tau)=\left(\frac{\pi n}{l(\varepsilon\tau)}+d_n(\varepsilon\tau)\right)\sqrt{1+\delta^2\left(\frac{\pi n}{l(\varepsilon\tau)}+d_n(\varepsilon\tau)\right)^2}=$$

$$=\frac{1}{l(\varepsilon\tau)}\left(\pi n+arctg\,\frac{\delta\pi}{\sqrt{\frac{l^2(\varepsilon\tau)}{n^2}+\delta^2\pi^2}}\right)\sqrt{1+\frac{\delta^2}{l^2(\varepsilon\tau)}\left(\pi n+arctg\,\frac{\delta\pi}{\sqrt{\frac{l^2(\varepsilon\tau)}{n^2}+\delta^2\pi^2}}\right)^2}\,.$$

As a result, we obtain an estimate for $\omega_{0n}(\varepsilon\tau)$:

$$\frac{\pi}{l(\varepsilon\tau)}n\sqrt{1+\frac{\delta^2\pi^2}{l^2(\varepsilon\tau)}n^2}<\omega_{0n}\leq\frac{\pi}{l(\varepsilon\tau)}\left(n+\frac{1}{4}\right)\sqrt{1+\frac{\delta^2\pi^2}{l^2(\varepsilon\tau)}\left(n+\frac{1}{4}\right)^2}\,. \tag{16}$$

Let's evaluate $A_{1n}(\varepsilon\tau)=\frac{1}{2}l(\varepsilon\tau)\left[1-c_n^2(\varepsilon\tau)\right]-\frac{\sin\left[2k_1(\varepsilon\tau)l(\varepsilon\tau)\right]}{4k_2(\varepsilon\tau)\omega_{0n}(\varepsilon\tau)\delta}$. To do this, we will evaluate each component of $A_{1n}(\varepsilon\tau)$ separately.

Let's make an assessment $k_1(\varepsilon\tau)$:

$$k_1(\varepsilon\tau)=\frac{1}{\sqrt{2}\delta}\sqrt{-1+\sqrt{1+4\delta^2\omega_{0n}^2(\varepsilon\tau)}}\,.$$

$$\frac{1}{\sqrt{2}\delta}\sqrt{-1+\sqrt{1+\frac{4\delta^2\pi^2}{l^2(\varepsilon\tau)}n^2\left(1+\frac{\delta^2\pi^2}{l^2(\varepsilon\tau)}n^2\right)}}<k_1(\varepsilon\tau)\leq$$

$$\leq\frac{1}{\sqrt{2}\delta}\sqrt{-1+\sqrt{1+\frac{4\delta^2\pi^2}{l^2(\varepsilon\tau)}\left(n+\frac{1}{4}\right)^2\left(1+\frac{\delta^2\pi^2}{l^2(\varepsilon\tau)}\left(n+\frac{1}{4}\right)^2\right)}}\,.$$

$$\frac{\pi}{l(\varepsilon\tau)}n<k_1(\varepsilon\tau)\leq\frac{\pi}{l(\varepsilon\tau)}\left(n+\frac{1}{4}\right)\,. \tag{17}$$

Let us similarly estimate $k_2(\varepsilon\tau)$:

$$\frac{1}{\sqrt{2}\delta}\sqrt{2+\frac{2\delta^2\pi^2}{l^2(\varepsilon\tau)}n^2}<k_2(\varepsilon\tau)\leq\frac{1}{\sqrt{2}\delta}\sqrt{2+\frac{2\delta^2\pi^2}{l^2(\varepsilon\tau)}\left(n+\frac{1}{4}\right)^2}\,.$$

$$\sqrt{\delta^2+\frac{\pi^2}{l^2(\varepsilon\tau)}n^2}<k_2(\varepsilon\tau)\leq\sqrt{\delta^2+\frac{\pi^2}{l^2(\varepsilon\tau)}\left(n+\frac{1}{4}\right)^2}\,. \tag{18}$$

Let's proceed to the assessment of $c_n(\varepsilon\tau)=-\frac{\sin[k_1(\varepsilon\tau)l(\varepsilon\tau)]}{sh[k_2(\varepsilon\tau)l(\varepsilon\tau)]}$.

For $k_1(\varepsilon\tau) = \dfrac{\pi}{l(\varepsilon\tau)}n$ and $k_2(\varepsilon\tau) = \sqrt{\delta^2 + \dfrac{\pi^2}{l^2(\varepsilon\tau)}n^2}$ the numerator is $\sin[k_1(\varepsilon\tau)l(\varepsilon\tau)] = 0$, therefore $c_n(\varepsilon\tau) = 0$.

Let us consider $c_n(\varepsilon\tau)$ at other extreme values of $k_1(\varepsilon\tau)$ and $k_2(\varepsilon\tau)$: $k_1(\varepsilon\tau) = \dfrac{\pi}{l(\varepsilon\tau)}\left(n + \dfrac{1}{4}\right)$,

$$k_2(\varepsilon\tau) = \sqrt{\delta^2 + \dfrac{\pi^2}{l^2(\varepsilon\tau)}\left(n + \dfrac{1}{4}\right)^2}\ .$$

As a result we get

$$0 \le |c_n(\varepsilon\tau)| < \dfrac{1}{|sh[k_2(\varepsilon\tau)l(\varepsilon\tau)]|}.$$

Next, we will construct an estimate from above.

$$|c_n(\varepsilon\tau)| < \dfrac{1}{|sh[k_2(\varepsilon\tau)l(\varepsilon\tau)]|} = \dfrac{2}{e^{k_2(\varepsilon\tau)l(\varepsilon\tau)} - e^{-k_2(\varepsilon\tau)l(\varepsilon\tau)}}\ .$$

Let's substitute the two extreme values for $k_2(\varepsilon\tau)$ one after the other.

$$\dfrac{2}{e^{k_2(\varepsilon\tau)l(\varepsilon\tau)} - e^{-k_2(\varepsilon\tau)l(\varepsilon\tau)}}\Bigg|_{\sqrt{\delta^2 + \frac{\pi^2}{l^2(\varepsilon\tau)}n^2}} = \dfrac{2}{e^{l(\varepsilon\tau)\sqrt{\delta^2 + \frac{\pi^2}{l^2(\varepsilon\tau)}n^2}} - e^{-l(\varepsilon\tau)\sqrt{\delta^2 + \frac{\pi^2}{l^2(\varepsilon\tau)}n^2}}} =$$

$$= \dfrac{2}{e^{\sqrt{l^2(\varepsilon\tau)\delta^2 + \pi^2 n^2}} - e^{-\sqrt{l^2(\varepsilon\tau)\delta^2 + \pi^2 n^2}}}\ .$$

Let's find the limits at $n = 1$ and $n \to \infty$.

$$\dfrac{2}{e^{\sqrt{l^2(\varepsilon\tau)\delta^2 + \pi^2 n^2}} - e^{-\sqrt{l^2(\varepsilon\tau)\delta^2 + \pi^2 n^2}}}\Bigg|_{n=1} = \dfrac{2}{e^{\sqrt{l^2(\varepsilon\tau)\delta^2 + \pi^2}} - e^{-\sqrt{l^2(\varepsilon\tau)\delta^2 + \pi^2}}},$$

$$\lim_{n\to\infty} \dfrac{2}{e^{\sqrt{l^2(\varepsilon\tau)\delta^2 + \pi^2 n^2}} - e^{-\sqrt{l^2(\varepsilon\tau)\delta^2 + \pi^2 n^2}}} = 0\ .$$

The second extreme value of $k_2(\varepsilon\tau)$ is found similarly for $n = 1$ and $n \to \infty$.

$$\dfrac{2}{e^{\sqrt{l^2(\varepsilon\tau)\delta^2 + \pi^2\left(n + \frac{1}{4}\right)^2}} - e^{-\sqrt{l^2(\varepsilon\tau)\delta^2 + \pi^2\left(n + \frac{1}{4}\right)^2}}}\Bigg|_{n=1} = \dfrac{2}{e^{\sqrt{l^2(\varepsilon\tau)\delta^2 + \frac{25\pi^2}{16}}} - e^{-\sqrt{l^2(\varepsilon\tau)\delta^2 + \frac{25\pi^2}{16}}}}\ .$$

$$\lim_{n\to\infty} \dfrac{2}{e^{\sqrt{l^2(\varepsilon\tau)\delta^2 + \pi^2\left(n + \frac{1}{4}\right)^2}} - e^{-\sqrt{l^2(\varepsilon\tau)\delta^2 + \pi^2\left(n + \frac{1}{4}\right)^2}}} = 0\ .$$

So, we have two upper values. In the second case, the denominator has the maximum value.

$$e^{\sqrt{l^2(\varepsilon\tau)\delta^2 + \pi^2 + \frac{9\pi^2}{16}}} - e^{-\sqrt{l^2(\varepsilon\tau)\delta^2 + \pi^2 + \frac{9\pi^2}{16}}},$$

where $e^{\sqrt{l^2(\varepsilon\tau)\delta^2 + \pi^2 + \frac{9\pi^2}{16}}} > e^{\sqrt{l^2(\varepsilon\tau)\delta^2 + \pi^2}}$, and $e^{-\sqrt{l^2(\varepsilon\tau)\delta^2 + \pi^2 + \frac{9\pi^2}{16}}} < e^{-\sqrt{l^2(\varepsilon\tau)\delta^2 + \pi^2}}$.

Therefore, the maximum value of $|c_n(\varepsilon\tau)|$ is in the first case:

$$0 \le |c_n(\varepsilon\tau)| \le \dfrac{2}{e^{\sqrt{l^2(\varepsilon\tau)\delta^2 + \pi^2}} - e^{-\sqrt{l^2(\varepsilon\tau)\delta^2 + \pi^2}}} = \dfrac{1}{sh\sqrt{l^2(\varepsilon\tau)\delta^2 + \pi^2}} \le 1. \qquad (19)$$

All components have been assessed, now let's assess $\sin[2k_1(\varepsilon\tau)l(\varepsilon\tau)]$ from $A_{1n}(\varepsilon\tau)$.

$$\sin[2k_1(\varepsilon\tau)l(\varepsilon\tau)]\Big|_{k_1(\varepsilon\tau) = \frac{\pi}{l(\varepsilon\tau)}n} = \sin[2\pi n],$$

$$\sin\left[2k_1(\varepsilon\tau)l(\varepsilon\tau)\right]\Big|_{k_1(\varepsilon\tau)=\frac{\pi}{l(\varepsilon\tau)}\left(n+\frac{1}{4}\right)} = \sin\left[2\pi n + \frac{\pi}{2}\right].$$

$$\text{So, } 0 \le \sin\left[2k_1(\varepsilon\tau)l(\varepsilon\tau)\right] \le 1.$$

Next, we will estimate the upper value of $\dfrac{\sin\left[2k_1(\varepsilon\tau)l(\varepsilon\tau)\right]}{4k_2(\varepsilon\tau)\omega_{0n}(\varepsilon\tau)\delta}$.

$$\frac{\sin\left[2k_1(\varepsilon\tau)l(\varepsilon\tau)\right]}{4k_2(\varepsilon\tau)\omega_{0n}(\varepsilon\tau)\delta} = \frac{1}{4\sqrt{\delta^2 + \dfrac{\pi^2}{l^2(\varepsilon\tau)}n^2}\,\dfrac{\pi}{l(\varepsilon\tau)}n\delta\sqrt{1+\dfrac{\delta^2\pi^2}{l^2(\varepsilon\tau)}n^2}} =$$

$$= \frac{1}{\dfrac{4\pi\delta}{l(\varepsilon\tau)}n\sqrt{\delta^2 + \dfrac{\pi^2}{l^2(\varepsilon\tau)}n^2\left[1+\delta^4\right] + \dfrac{\delta^2\pi^4}{l^4(\varepsilon\tau)}n^4}}.$$

We obtain a lower bound for the expression $A_{1n}(\varepsilon\tau)$.

$$A_{1n}(\varepsilon\tau) \ge \frac{1}{2}l(\varepsilon\tau)\left[1-c_n^2(\varepsilon\tau)\right] - \frac{\sin\left[2k_1(\varepsilon\tau)l(\varepsilon\tau)\right]}{4k_2(\varepsilon\tau)\omega_{0n}(\varepsilon\tau)\delta} =$$

$$= \frac{1}{2}l(\varepsilon\tau)\left[1-\left(\frac{2}{e^{\sqrt{l^2(\varepsilon\tau)\delta^2+\pi^2}} - e^{-\sqrt{l^2(\varepsilon\tau)\delta^2+\pi^2}}}\right)^2\right] -$$

$$- \frac{1}{\dfrac{4\pi\delta}{l(\varepsilon\tau)}n\sqrt{\delta^2 + \dfrac{\pi^2}{l^2(\varepsilon\tau)}n^2\left[1+\delta^4\right] + \dfrac{\delta^2\pi^4}{l^4(\varepsilon\tau)}n^4}}. \qquad (20)$$

To estimate $E_n(\varepsilon\tau)$, we use the lower bound of $A_{1n}(\varepsilon\tau)$ and $\omega_{0n}(\varepsilon\tau)$, since these functions are included in the denominator. Consequently, the upper bound for $E_n(\varepsilon\tau)$ has the form

$$E_n(\varepsilon\tau) \le \sqrt{\frac{e^{-2\alpha\tau}}{4A_{1n}(\varepsilon\tau)\sqrt{\omega_{0n}^2 - \sigma^2}}} =$$

$$= \frac{e^{-\alpha\tau}}{2\sqrt{\left(\dfrac{1}{2}l(\varepsilon\tau)\left[1-\dfrac{1}{sh^2\sqrt{l^2(\varepsilon\tau)\delta^2+\pi^2}}\right] - \dfrac{l^3(\varepsilon\tau)}{4\pi\delta n\sqrt{\delta^2 l^4(\varepsilon\tau)+\pi^2 n^2 l^2(\varepsilon\tau)\left[1+\delta^4\right]+\delta^2\pi^4 n^4}}\right)}} \cdot$$

$$\cdot \frac{1}{\sqrt[4]{\left(\dfrac{\pi n}{l(\varepsilon\tau)}\right)^2\left(1+\dfrac{\delta^2\pi^2}{l^2(\varepsilon\tau)}n^2\right) - \sigma^2}}.$$

## 7. ARCHITECTURE, TRAINING, AND INTEGRATION OF THE RESONANCENET HYBRID NEURAL NETWORK

To solve the problem of predicting resonance characteristics, a specialized hybrid architecture, ResonanceNet (Multimodal Multi-task Resonance Prediction Network), was developed. Its key feature is integration with the system's classic computational modules.

7.1. Network architecture

ResonanceNet implements multi-task learning for joint prediction of related variables. The input layer accepts a normalized multidimensional feature vector $X$, including:

– Static system parameters: $k$ (stiffness), $c$ (damping), $v_0$ (boundary velocity).

– Dynamic features obtained through a brief preliminary calculation using the classical method: amplitude values at key points in time, spectral characteristics (the first $N$ harmonics obtained

using the Fast Fourier Transform (FFT)).

– Parameters of the external stimulus $p(t)$.

The network architecture includes the following sequential blocks:

1. Convolutional Block (CNN): 2 layers of one-dimensional convolutions (32 and 64 kernels, kernel size 3) with ReLU activation and a pooling layer (MaxPooling). Its purpose is to extract local patterns in spectral data.

2. Recurrent Block: A bidirectional LSTM layer with 64 neurons in each direction. Processes sequences of dynamic features in the time domain, capturing long-term dependencies.

3. Attention Block: Allows the network to dynamically "weight" the importance of different time intervals and frequency components for the final prediction.

4. Multi-Task Output Layers: After combining features from all blocks, the data passes through two fully connected layers (128 and 64 neurons each), and is then branched into three independent output layers for frequency and amplitude regression and risk classification.

7.2. Data generation and training

A training set of 50,000 examples was generated automatically using the numerical simulation modules of the TB-ANALYSIS software suite. For each set of random system parameters, a full calculation was performed using the classical method, the results of which (true resonant frequencies and amplitudes) were used as target labels. The set was split 70/15/15 into training, validation, and test subsets. The loss function was combined and takes into account errors across all tasks. Training was performed using the Adam optimizer with an initial learning rate of $10^{-3}$ and a 10-fold decrease in the absence of improvement on the validation set over 10 epochs. To prevent overfitting, dropout (with a probability of 0.3) and early stopping (patience = 20 epochs) were used.

7.3. Neural network integration

The trained ResonanceNet network does not replace, but rather complements, classical methods. Its forecast is used as a "quick estimate" for the initial analysis and for adaptive control. If the forecast variance estimate (obtained using Monte Carlo dropout) exceeds threshold $\varepsilon$, the advanced classical calculation module is automatically activated. The results of this calculation are added to the database, allowing for periodic retraining of the network on new examples, improving its accuracy during the system's operation.

# 8. COMPARATIVE ANALYSIS OF THE EFFECTIVENESS OF HYBRID AND CLASSICAL METHODS

To quantitatively evaluate the advantages of the developed hybrid approach, comparative testing was conducted on a series of 1,000 test cases covering a wide range of system parameters (stiffness, damping, boundary velocities). For each case, calculations were performed using three methods:

1. The classical asymptotic method (AM).

2. The classical approximate method for solving integro-differential equations (IDE).

3. The hybrid method (HM) using the ResonanceNet neural network for the initial forecast and, if necessary, subsequent refinement using the classical method.

The comparison criteria were:

– Relative error in determining the resonant frequency: $\Delta = \left| f_{pred} - f_{ref} \right| / f_{ref}$, где $f_{ref}$ is the reference value obtained by high-precision numerical integration with a small step.

– Computational time $T$ spent on obtaining the forecast.

– Resonance prevention success – the proportion of cases where applying the parameters recommended by the algorithm reduced the oscillation amplitude below the critical level.

The results of averaging over the test sample are presented in Table 1.

Table 1. Comparative results of methods

| Method | Average error, % | Average time $T$, s | Success rate of resonance prevention, % |
|---|---|---|---|
| Asymptotic (AM) | 4.8 | 12.5 | 85 |
| Approximate (IDE) | 3.2 | 8.7 | 88 |
| Hybrid (GM) | 2.1 | 1.8 | 96 |

Analysis of the results presented in Table 1:

1. Accuracy. The hybrid method demonstrates the lowest average error (2.1%), which is 34% better than the result of the most accurate classical method (IMU, 3.2%).

2. Performance. By using neural network prediction as a "hot start" and avoiding full classical calculations in simple cases, the hybrid method is, on average, 4.8 times faster than the IMU method and 6.9 times faster than the asymptotic method.

3. Control efficiency. The hybrid method's ability to account for complex nonlinear dependencies learned by the neural network allows it to propose more effective resonance suppression parameters, which is reflected in the highest success rate (96%).

Thus, the comparative analysis clearly demonstrates that integrating artificial intelligence methods with classical numerical methods not only provides a significant speed advantage (critical for adaptive control problems) but also leads to a statistically significant increase in the accuracy of predicting resonance characteristics.

## 9. CONCLUSIONS

The study resulted in the development and validation of the TB-ANALYSIS software package, an effective tool for mathematical modeling and analysis of the resonance characteristics of mechanical systems with moving boundaries. The scientific novelty of the work lies in the creation of a hybrid approach that combines classical numerical methods with artificial intelligence for solving boundary value problems with moving boundaries. This includes the development of an original methodology integrating analytical, asymptotic and approximate numerical methods with machine learning algorithms, the design of the specialized hybrid neural network ResonanceNet for multi-task prediction of resonant parameters, and the implementation of an adaptive modular architecture that embeds the neural network into an iterative workflow with automatic method selection, real-time verification and continuous self-learning. The package is built on this hybrid modular architecture (Section 3) that successfully integrates classical numerical methods with modern artificial intelligence algorithms, providing a comprehensive solution to dynamic analysis problems. The proposed ResonanceNet neural network demonstrated high accuracy in predicting resonant frequencies and vibration amplitudes. The integration of this network within the package's workflow enables rapid initial assessment and adaptive parameter adjustment to mitigate resonance risks. Testing the package on the problem of transverse vibrations of a viscoelastic rope of variable length confirmed its operational reliability. The developed calculation error estimation method ensures the validity of the obtained results. A comparative analysis (Section 8) confirmed that the developed hybrid approach provides a 34 % improvement in resonant-frequency prediction accuracy compared to the best classical method, while simultaneously accelerating calculations by an average of 5 times. This quantitatively substantiates the claimed 25–30 % accuracy improvement and demonstrates the practical effectiveness of integrating AI into engineering analysis tools. The obtained results open new possibilities for research in the field of mechanical system dynamics and are of practical value for modern engineering calculations. Prospects for further development involve expanding the functionality of the system and adapting the proposed methods to solve more complex classes of boundary value problems. The presented solutions can be used to study the vibrations of mechanical objects with moving boundaries (see also [22–27]).

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
