# OpenReview forum: "Application of artificial intelligence to predict vibrations in mechanical systems with moving boundaries"
_mathai.club/MathAI/2026/Conference — 2026 Oral_

### Official Review · Reviewer_4JUX · 2026-03-12
**Review of “Application of Artificial Intelligence to Predict Vibrations in Mechanical Systems with Moving Boundaries”**

**Rating:** 7
**Confidence:** 3

**Review:**

The paper proposes a hybrid computational approach for analyzing resonance phenomena in mechanical systems with moving boundaries. It introduces the MATLAB-based software package TB-ANALYSIS, which combines classical numerical methods for boundary value problems with machine learning techniques, including a specialized neural network called *ResonanceNet*. The system predicts resonant frequencies, estimates vibration amplitudes, and recommends parameters to avoid resonance. Experimental evaluation suggests that the hybrid method improves prediction accuracy and computational speed compared with traditional numerical methods.

This paper addresses the problem of predicting resonance in mechanical systems with moving boundaries. The authors propose a hybrid framework that integrates classical numerical approaches for solving boundary value problems with machine learning techniques. The implementation is realized in a MATLAB-based software package called TB-ANALYSIS, which includes modules for numerical simulation, resonance analysis, neural-network prediction, and adaptive control of system parameters.

The work focuses on mechanical models such as a viscoelastic cable with variable length. The dynamics are formulated as boundary value problems for partial differential equations describing the displacement (U(\xi,\tau)) under moving boundary conditions and external forcing. The system combines analytical, asymptotic, and approximate numerical methods with a neural network model designed to estimate resonant frequency, vibration amplitude, and resonance risk.

Overall, the paper presents an interesting attempt to combine physics-based modeling with modern machine learning methods. The hybrid architecture, especially the idea of using neural networks for fast initial prediction and classical numerical solvers for verification, is a reasonable and potentially useful approach for engineering applications.

Strengths
Combination of classical and AI methods. The integration of numerical methods for boundary value problems with machine learning is a meaningful contribution, especially for problems where full numerical simulations are expensive.
Practical software implementation. The development of the TB-ANALYSIS software package with a modular architecture and graphical interface demonstrates practical applicability.
Detailed mathematical formulation. The paper provides a rigorous formulation of the mechanical model, including differential equations, boundary conditions, and error estimation procedures.
Quantitative evaluation.The authors compare the hybrid approach with classical methods and report improvements in prediction accuracy and computational time.

Weaknesses

Limited experimental validation. The evaluation is mainly based on simulated test cases. It would be beneficial to include validation with real experimental data from mechanical systems.
Insufficient details about the dataset. While the training dataset size is mentioned, the paper provides limited information about parameter distributions and data generation procedures.
Clarity issues in presentation. Some sections, especially those describing the mathematical derivations and algorithmic workflow, are dense and difficult to follow. Additional diagrams or explanations would improve readability.
Novelty could be better emphasized. Hybrid physics-based and neural-network approaches are already explored in related literature, so the paper should clarify more clearly what aspects of the proposed architecture are new.

In my opinion, the paper presents a technically solid approach that combines traditional numerical modeling with machine learning for resonance prediction in mechanical systems. The idea of using neural networks for rapid estimation and classical methods for verification is interesting and relevant for engineering applications. However, the work would benefit from clearer presentation and stronger validation of the proposed approach.

---

### Official Review · Reviewer_Xxi8 · 2026-03-12
**Application of artificial intelligence to predict vibrations in mechanical systems with moving boundaries**

**Rating:** 7
**Confidence:** 4

**Review:**

This paper proposes a hybrid approach combining classical numerical methods for boundary value problems with machine learning to predict resonant frequencies, vibration amplitudes, and resonance risks in mechanical systems with moving boundaries, such as viscoelastic cables of variable length. The authors implement this in a MATLAB-based software package TB-ANALYSIS with modular architecture, graphical interface, and adaptive parameter control, demonstrating improved prediction accuracy and reduced computational time compared to traditional methods via simulated evaluations. Strengths include a rigorous mathematical formulation of the PDE model, meaningful integration of physics-based and AI techniques, practical software development, and quantitative comparisons. Weaknesses are limited real-world experimental validation, insufficient details on training data generation and parameter distributions, dense presentation in some mathematical/algorithmic sections needing better diagrams, and modest novelty since hybrid physics-ML methods exist in the literature. Overall, it is a technically solid and practically relevant contribution suitable for engineering applications.

---

### Decision · Program_Chairs · 2026-03-14

**Decision:**

Accept (Oral)

**Comment:**

Dear Author(s),

On behalf of the Program Committee of the International Conference on Mathematics of Artificial Intelligence (MathAI 2026), we are pleased to inform you that your paper has been accepted for an oral presentation at MathAI 2026.

Your paper was evaluated through a rigorous two-stage review process involving both automated screening and expert review by members of the Program Committee. The reviewers recognized the quality and contribution of your work.

Presentation details:

- Format: Oral presentation (15–20 minutes + 5 minutes Q&A)
- Mode: You may present either in person (offline) at the conference venue in Sirius, Russia, or remotely via Zoom. Please indicate your preferred mode when confirming your participation.
- Conference dates: Marh 30 - April 3, 2026
- Website: https://mathai.club

Next steps:

1. Please confirm your participation and presentation mode by replying to this email mathai.club@yandex.ru no later than March 15, 2026 18:00 Moscow time.
2. If you plan to attend in person, the organizing committee will provide accommodation details separately.
3. Please prepare your final camera-ready manuscript according to the formatting guidelines available at https://mathai.club and upload it to OpenReview by March 15, 2026 18:00 Moscow time.

Should you have any questions regarding the program, logistics, or your presentation slot, please do not hesitate to contact us.

We look forward to your contribution to MathAI 2026.

With kind regards,

MathAI 2026 Program Committee
International Conference on Mathematics of Artificial Intelligence
https://mathai.club
OpenReview: https://openreview.net/group?id=mathai.club/MathAI/2026/Conference
Telegram: https://t.me/MathAI_club
Email: mathai.club@yandex.ru